# Self-assembly directed one-step synthesis of [4]radialene on Cu(100) surfaces

Qing Li[1], Jianzhi Gao[2], Youyong Li[1], Miguel Fuentes-Cabrera[3,4], Mengxi Liu[5], Xiaohui Qiu[5,6], Haiping Lin[1], Lifeng Chi[1] & Minghu Pan[7]

The synthetic challenges of radialenes have precluded their practical applications. Here, we report a one-step synthetic protocol of [4]radialene on a copper surface. High-resolution scanning tunneling microscopy measurements reveal that such catalytic reaction proceeds readily with high selectivity at the temperature below 120 K. First-principles calculations show that the reaction pathway is characterized by firstly the cooperative inter-molecular hydrogen tautomerization and then the C–C bond formation. The feasibility of such cyclotetramerization reaction can be interpreted by the surface effect of Cu(100), which firstly plays an important role in directing the molecular assembly and then serves as an active catalyst in the hydrogen tautomerization and C–C coupling processes. This work presents not only a novel strategy to the scant number of synthetic methods to produce [4]radialenes via a novel [1 + 1 + 1 + 1] reaction pathway, but also a successful example of C–C bond coupling reactions guided by the surface-induced C–H/π assembly.

[1] Institute of Functional Nano & Soft Materials (FUNSOM), Jiangsu Key Laboratory for Carbon-Based Functional Materials & Devices, Soochow University, Suzhou 215123, China. [2] School of Physics and Information Technology, Shaanxi Normal University, Xi'an 710062, China. [3] Center for Nanophase Materials Sciences, Oak Ridge National Laboratory, Oak Ridge, TN 37831, USA. [4] Computational Sciences and Engineering Division, Oak Ridge National Laboratory, Oak Ridge, TN 37831, USA. [5] CAS Key Laboratory of Standardization and Measurement for Nanotechnology, CAS Center for Excellence in Nanoscience, National Center for Nanoscience and Technology, Beijing 100190, China. [6] University of Chinese Academy of Sciences, Beijing 100049, China. [7] School of Physics, Huazhong University of Science and Technology, Wuhan 430074, China. Correspondence and requests for materials should be addressed to H.L. (email: hplin@suda.edu.cn) or to L.C. (email: chilf@suda.edu.cn) or to M.P. (email: minghupan@hust.edu.cn)

The ultimate goal of synthetic chemistry is to obtain the desired molecular structures from readily available precursors with rationally designed step-economic synthetic strategies. In this regard, the constructions of n-membered cyclic carbon scaffolds have been an attractive research hotspot, owing to their significant importance in fine-chemical and pharmaceutical industries. Among the family of ring structures, the production of tetramethylenecyclobutane ([4]radialenes) and their derivatives have attracted extensive theoretical and synthetic interest due to their special cross-conjugated eight-center, eight π-electron systems, which have been reported to show interesting properties such as multiredox potential[1]. The successful synthesis of [4]radialene was firstly reported by Griffin and Peterson[2]. Over the past a few decades, the [4]radialenes and their derivatives have been synthesized via two major strategies: (i) the cyclization of precursor olefins (e.g., cumulenes) to form the radialene core[3,4] and (ii) instruction of the exocyclic olefin groups after the construction of the four-membered carbon skeleton[5]. Although tremendous efforts have been devoted to access these compounds, owing to the drastic reaction coordinates, even the most successful synthetic methods are limited to the poor yields and selectivity of the high symmetric cores of [4]radialenes[6].

Ideally, the most step-economic and straight forward synthetic strategy to form [4]radialenes is the [1 + 1 + 1 + 1] cyclotetramerization of alkyne groups (as shown in Fig. 1), in which the bond formations and ring constructions of four-membered cyclic carbon scaffolds take place in just one step via the hydrogen tautomerization. Despite of the appeared simplicity, successful examples of such a synthetic route has not yet been reported, because of three practical challenges: (1) the distances and orientations of the four alkyne groups must be well matched and stabilized, prior to the tetramerization reaction. This requirement is intrinsically unrealistic in gases or liquids, but can possibly be achieved on solid surfaces by means of molecular assembly. (2) The activation energy barrier of the tetramerization reaction should be close to, if not less than, the energy cost to break the self-assembly of monomers, so that the cycloaddition proceeds immediately when the molecular assembly is formed. (3) The hydrogen tautomerization and the formation of the four-membered carbon ring must take place almost at the same time. Otherwise, according to the rate of reaction, the formation of dimers or chain-like polymers would be kinetically much more favorable than that of tetramers. Thus, to selectively obtain [4]radialenes through the proposed [1 + 1 + 1 + 1] mechanism, the three challenging conditions (self-assembled reaction precursors, activation barrier and reaction pathway) must be spontaneously fulfilled by employing a rationally designed chemical environment.

Recent developments of "on-surface chemistry" have shown that surfaces may provide effective controlling parameters to the

reaction mechanisms of chemical reactions[7–21]. With the assistance of catalytic surfaces, various one-dimensional (1D) and two-dimensional (2D) polymer structures were constructed on metal surfaces[22–30]. In these reactions, metal surfaces serve not only as catalysts, but also as 2D templates to selectively tune the in-plane self-assembly of the reactants. For example, Zhang et al. reported that the C–C bond coupling between reaction precursors can be effectively guided by the molecular self-assembly, so that the chirality can be transferred from the assembly structure to the oligomer reaction products[31]. Our previous studies have shown that acetyls derivatives may self-assemble into magic supramolecular structures[32]. In addition, the C–C triple bonds are fairly reactive on metal surfaces[12,33–36]. Thus, correct combinations of surface electronic states, symmetries, and lattices of transition metals may provide possibilities to overcome the above-mentioned three challenges for the proposed [1 + 1 + 1 + 1] synthetic route.

By using phenylacetylene as the precursor, here we successfully achieved the selective synthesis of the tetraphenyl[4]radialenes on Cu(100) surfaces. The tetramerized products were visualized by high-resolution scanning tunneling microscopy (STM), and then further evidenced by tip manipulations. Combining with density functional theory (DFT) calculations, we elucidated that the importance of the Cu(100) surfaces is twofold: (i) guiding the self-assembly of phenylacetylene molecules to form a tetramer-oriented adsorption configuration, in which the attractive force between the monomers is the C–H/π interaction and (ii) working as an active catalyst for the cycloaddition reactions. Detailed examinations of the reaction pathway show that such a cyclization of four phenylacetylene molecules to form [4]radialenes was proceeding via an unusual synergic inter-molecular hydrogen tautomerization and inter-molecular C–C bond formation process. This work demonstrates not only a feasible synthetic method to selectively produce [4]radialenes, but also opens a new avenue to C–C coupling guided by the surface-induced C–H/π assembly.

## Results

**Tetramer products formation.** We firstly dose the phenylacetylene molecules on a Cu(100) substrate with the substrate held at very low temperature (20 K) to identify their pristine adsorption configurations. Figure 2a gives a representative STM image after adsorbing 0.003 ML phenylacetylene molecules on Cu(100). The phenylacetylene monomers can be identified as the isolated bright protrusions. As seen, the monomers did not aggregate together, indicating the suppression of the molecular diffusions at low temperature on Cu(100) surfaces. The diffusion barrier of a pristine phenylacetylene monomer on a Cu(100) surface is 0.28 eV. Distinct structural evolutions take place after annealing the sample at 200 K for 20 min. As shown in Fig. 2b, in addition to the individual monomers, a number of tetramers were observed dispersing on the Cu(100) surface. Such STM features resemble to self-assembly structure of phenylacetylene molecules on the Au (111) surface, on which six phenylacetylene monomers were attracted by the weak C–H/π interactions to form hexamers[32].

Despite the apparent similarity, the tetramers observed in this work cannot be attributed to the molecular assembly of four phenylacetylene molecules. Firstly, acetylenes are known to be very active on copper surfaces, and have been reported to undergo Glaser coupling reactions[12,35,36] or the C–Cu bond formation with copper surfaces[29,33,34]. In particular, our previous studies have indicated that depositing phenylacetylene molecules onto the same surface (Cu(100)) held at 120 K leads to the formation of 1D polymers[34]. Thermodynamically, one should not obtain self-assembly structures after annealing the phenylacetylene decorated Cu(100) surfaces at a temperature much higher

**Fig. 1** The [1 + 1 + 1 + 1] cyclotetramerization reaction of self-assembled alkyne groups. The one-step synthesis of tetraphenyl[4]radialenes via a novel [1 + 1 + 1 + 1] cyclotetramerization reaction pathway on the Cu(100) surface

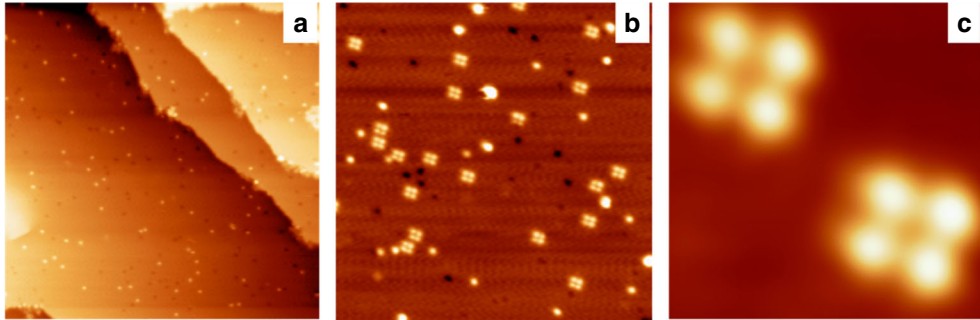

**Fig. 2** Formation of tetramers on Cu(100). **a** The STM image after dosing 0.003 ML phenylacetylene molecules on a Cu(100) surface held at 20 K. **b** The STM image after annealing the sample at 200 K for 20 min. **c** The high-resolution details of the tetramer features. The image size are 100 nm × 100 nm for **a**, 40 nm × 40 nm for **b**, and 4 nm × 4 nm for **c**. The scanning parameters are $V_b = 1$ V, $I_t = 50$ pA for all the images

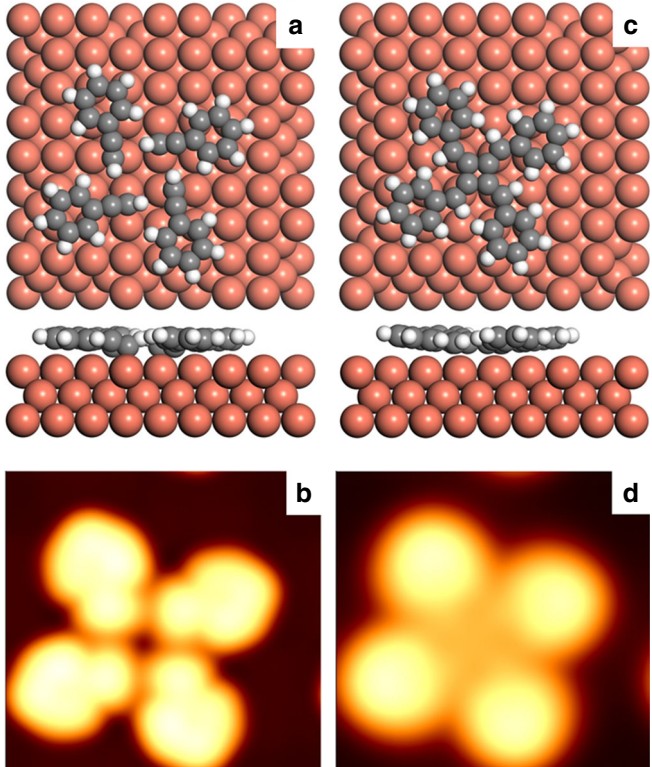

**Fig. 3** Optimized structure and STM simulation of self-assembled and covalent connected products. **a**, **b** The optimized structure and the corresponding simulated STM image ($V_b = 2.0$ V) of the molecular assembly of four phenylacetylene molecules on the Cu(100) surface. **c**, **d** The optimized structure and the corresponding simulated STM image ($V_b = 2.0$ V) of the covalently bonded tetramer on the Cu(100) surface. Color code: Cu (brown), C (gray), and H (white)

than 120 K. Secondly, DFT calculations and STM simulations were carried out to determine the chemical identity of the tetramers. Figure 3a shows the relaxed structure of molecular assembly of four phenylacetylene molecules. Similar to the molecular self-assembly on Au(111) surfaces[32], the phenylacetylene molecules were attracted by weak C–H/π interactions (0.12 eV per molecule from DFT calculations). The simulated STM image according to the optimized structure of self-assembly is shown in Fig. 3b. The phenylacetylene monomers are imaged as elongated protrusions, which are similar to the monomer in the self-assembled hexamer on Au(111)[32]. However, the simulated images are rather different from that observed on Cu(100) experimentally (Fig. 2b).

We therefore propose the experimentally observed tetramers to the tetraphenyl[4]radialene molecule formed via the cycloaddition reactions (Fig. 3c). As shown, the terminal hydrogen atom of each phenylacetylene molecule was transferred to the carbon atoms that directly connected with the phenyl groups. The resulted carbon radicals combine to each other to form the four-membered carbon ring of the tetraphenyl[4]radialene. The calculated STM images show that the phenyl groups are imaged as spherical protrusions, while the four-membered carbon ring appears as a depression, the distance between the centers of side-protrusions is about 8.0 Å. All the features are in excellent agreement with the experimental results. Note that the distance between the centers of side-protrusions is about 9.0 Å for the C–H/π interactions induced self-assembled tetramers (see Supplementary Figure 1 for details).

More interestingly, additional features in the center of the tetramers become visible at lower bias voltages (Fig. 4a). Take the topographic image acquired at −100 meV for example (Fig. 4b), one can observe four bright protrusions in the center of a tetramer, arranged in a square shape. The side distance of the protrusions is as small as 2.0 Å, which is close to but slightly larger than the length of the carbon–carbon single bond. The corresponding STM simulation at the −100 meV is shown in Fig. 4c. As seen, the calculated protrusion–protrusion distance is around 1.60 Å. The small difference between the theoretical and experimental result could be interpreted by the tip–molecule interaction (see Supplementary Figures 2 and 3 for more information). Both the size and the overall shape of the protrusions in the STM image fit well with the electronic states of the central four-member carbon ring of the proposed [4]radialene, as shown in Fig. 4c. In addition, we have considered alternative candidates for the tetramer structures that are formed via non-covalent interactions (see Supplementary Figure 4 for details). None of them can lead to four protrusions in the supramolecular center with such a small size. All these analysis then strongly suggested the validity of the synthesis of [4]radialene products on the copper surface.

**DFT and climbing image nudged elastic band calculations.** Subsequent climbing image nudged elastic band (CI-NEB) calculations indicate that such cyclotetramerization reaction proceeds via an unusual [1 + 1 + 1 + 1] mechanism. Figure 5 shows the energy profile and the structural model at each stage. The energy barrier for the initial state (IS) to the first transition state (TS1) is 1.28 eV (0.32 eV per molecule). In this transition state, the phenylacetylene molecules move towards each other. Of more importance, the alkyne groups are twisted so that their hydrogen atoms are close to the carbon atoms of the nearest molecules (the C–H distance is 1.61 Å). Crossing over the first transition state, the system reaches a meta-state (MS), at which the terminal

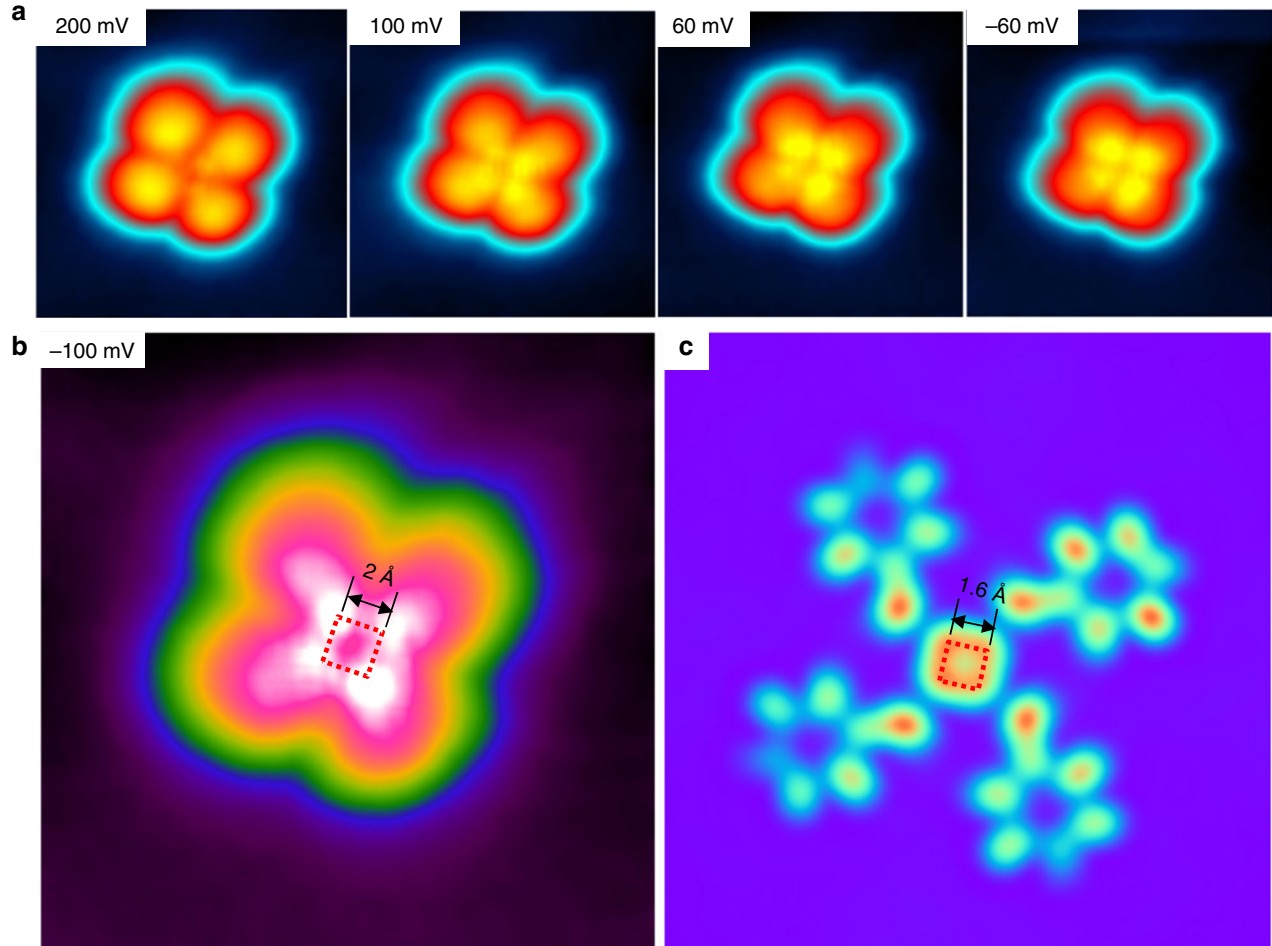

**Fig. 4** STM image at lower bias voltages. **a** The STM topographic images of the tetramer products measured at 200 mV, 100 mV, 60 mV, and −60 meV, respectively. **b** The STM topographic images of the tetramer product measured at −100 mV. The square shows the size of the central four protrusions. **c** The simulated STM image of a tetramer product. All the STM topographic images were obtained with $I_t = 100$ pA

hydrogen atoms of the alkyne groups have been transferred to the carbon atoms of their nearest neighbors. Each of the resulted carbon radicals is bonded to two copper atoms of the Cu(100) surface to form a meta-stable molecular complex. From this meta-state to the final state (TS2), the energy barrier is only 0.18 eV. Due to the formation of four C–C bonds in the last step, the whole cycloaddition reaction is an exothermic process (the overall energy drop is 4.67 eV).

Combing the STM observation with the theoretical calculations, the experimental observations can be completely interpreted. At the temperature of 20 K, the adsorbed phenylacetylene molecules were frozen on the copper surface because the thermal activation barrier of molecular diffusion is 0.28 eV. When the surface is heated gradually, the adsorbed molecules start to move around above the surface, and result in the self-assembled configuration as shown in Fig. 3a. The driving force of such molecular assembly is the non-covalent C–H/π interaction. As a result, the total energy is lower by 0.12 eV per molecule. Once such molecular-assembled structures are achieved, however, the subsequent [1 + 1 + 1 + 1] cyclotetramerization takes place immediately to produce the tetraphenyl[4]radialene, because the activation barrier for cycload-dition reaction is only 0.32 eV per molecule (0.04 eV higher than the energy barrier for molecular diffusion). Correspondingly, the molecular-assembled structures were not observed experimentally because of their short lifetime.

The above analysis reveals that the Cu(100) surface plays a key role in the cyclization reactions, particularly in guiding the

formation of the self-assembly intermediate state prior to the tetramerization reaction. This tells why the [1 + 1 + 1 + 1] cyclotetramerization of alkyne groups is hardly achieved in gases or solutions. To further emphasize this point, controlling experiments were performed by dosing phenylacetylene mole-cules onto the Cu(100) surface at a higher temperature (120 K). At this temperature, the thermal energy is higher than the energy requirement both to break the molecular assembly and to trigger the acetyls reactions on Cu(100). Correspondingly, the molecular-assembled tetramer cannot be formed before the occurrence of the reactions, prohibiting the cooperative inter-molecular hydro-gen tautomerization. As a consequence, the C–C coupling reactions occur between phenylacetylene molecules and result in the formation of chain-like polymers[34]. This controlling experiment has demonstrated a novel example about how to artificially guide the reaction mechanisms by rationally control-ling the reaction dynamics.

**Tip manipulations.** To further confirm the bonding nature of the experimentally observed tetramers, we carried out delicate tip manipulation experiments, which is a common and effective method to identify the strength of the interactions among the monomers in complex structures[11,37]. Figure 6a gives a repre-sentative STM topographic image, in which three tetramers were observed. To conduct the tip manipulation, we applied a negative voltage pulse of −3.2 V between STM tip and molecules (details

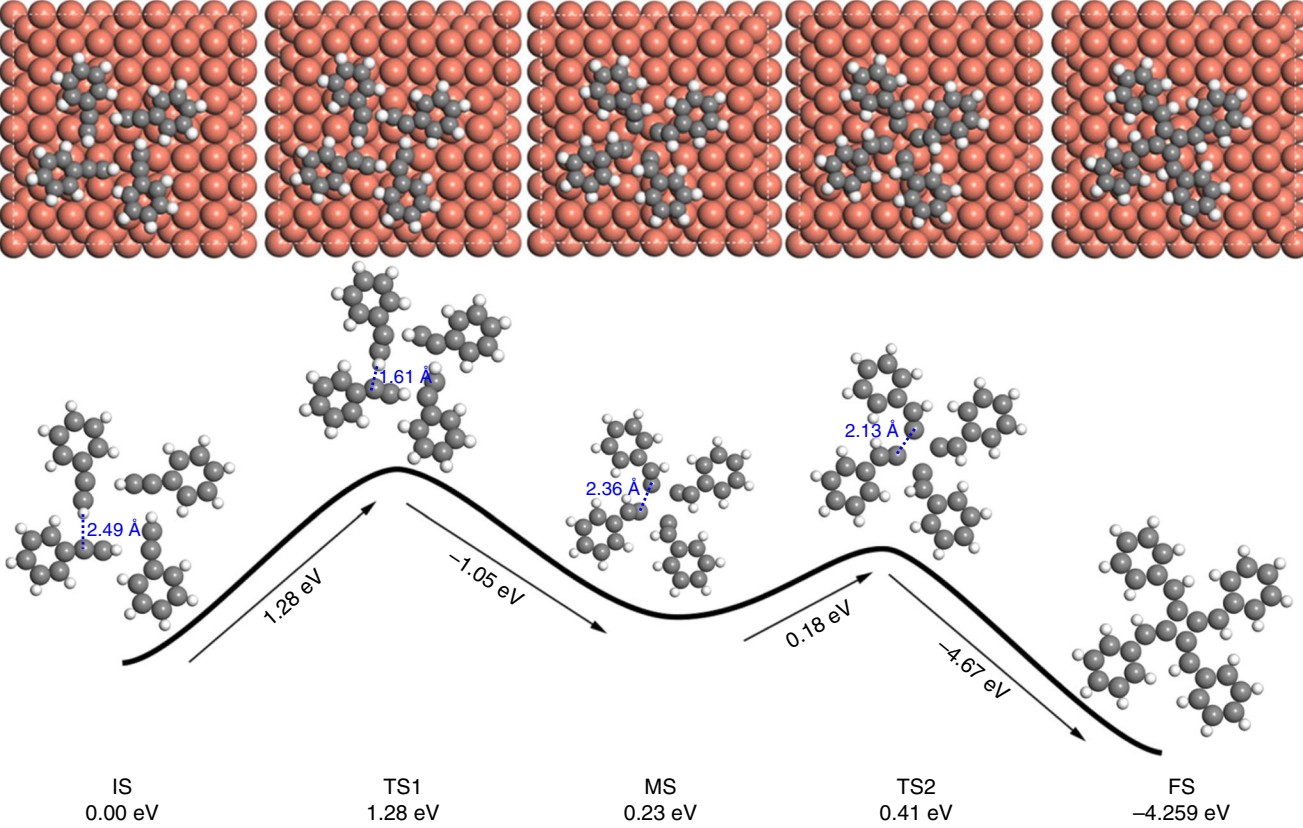

**Fig. 5** DFT calculations of the reaction pathway with corresponding energy profiles. The cyclotetramerization reaction proceeds via the cooperative intermolecular hydrogen tautomerization and a [1+1+1+1] cycloaddition mechanism

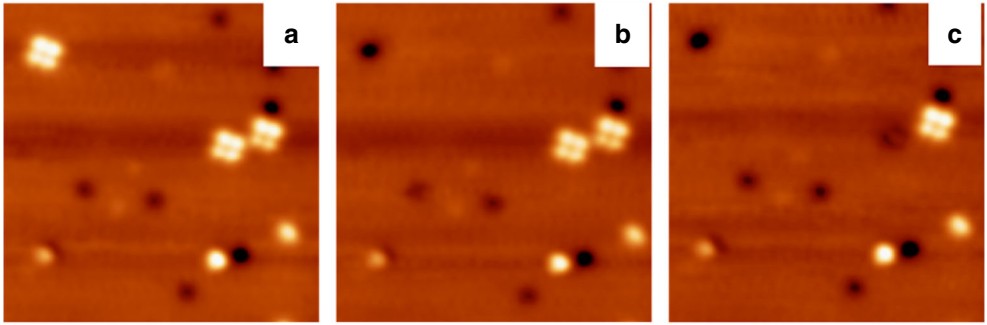

**Fig. 6** Controllable successive tip manipulation experiments. **a** A STM topographic image includes three tetramers. **b** After a pulse is applied, the tetramer located at the top left corner disappears. **c** One more tetramer disappears upon further tip pulsing. To conduct the tip manipulation, a voltage pulse of −3.2 V between STM tip and molecules was applied. The image sizes are 22 nm × 22 nm and the scanning parameters are $V_b = 1$ V, $I_t = 20$ pA for all the images

of the manipulation methods can be found in ref. [32]). We choose a region where three tetramers are observed (Fig. 6a). After the pulse is applied, the tetramer located at the top left corner disappears (Fig. 6b). Further, tip manipulation results in that the STM tip picks up one more tetramer, as seen in Fig. 6c. This phenomenon clearly indicates that each tetramer structure is composed of four monomers which are connected by strong attractive interactions, e.g., the covalent bonds. By contrast, on the Au(111) surface, we have prepared phenylacetylene hexamers, which are known to be formed by six pristine phenylacetylene molecules via non-covalent C–H/π interactions. By means of the same tip manipulation method, it is shown that only one phenylacetylene monomer is picked up by the STM tip (see Supporting information of ref. [32] for details). The different results of tip manipulation analysis for hexamers on Au(111)[32] and

tetramers on Cu(100) then suggest that the tetramers are not self-assembly structures, but the reaction products of cycloaddition.

**Spectra measurements**. Finally, we performed the d$Z$/d$V$ (represents the image potential, and related to the local density of states[38,39]) and d$I$/d$V$ measurements for the clean Cu(100) surface, single PA molecule, and the tetramer (Fig. 7). d$Z$/d$V$ spectroscopy (Fig. 7a) shows the Stark-shifted, first image-potential state associated with phenylacetylene monomer and tetraphenyl [4]radialene, compared to bare Cu(100) surface. It is believed that a molecular dipole moment induced by molecular adsorption lowers the surface work function[40,41], which results in the observed shift of image-potential state. For isolated tetraphenyl[4] radialene molecule, it induces 0.08 eV more energy downshift than that of isolated PA monomers, suggesting a stronger dipole

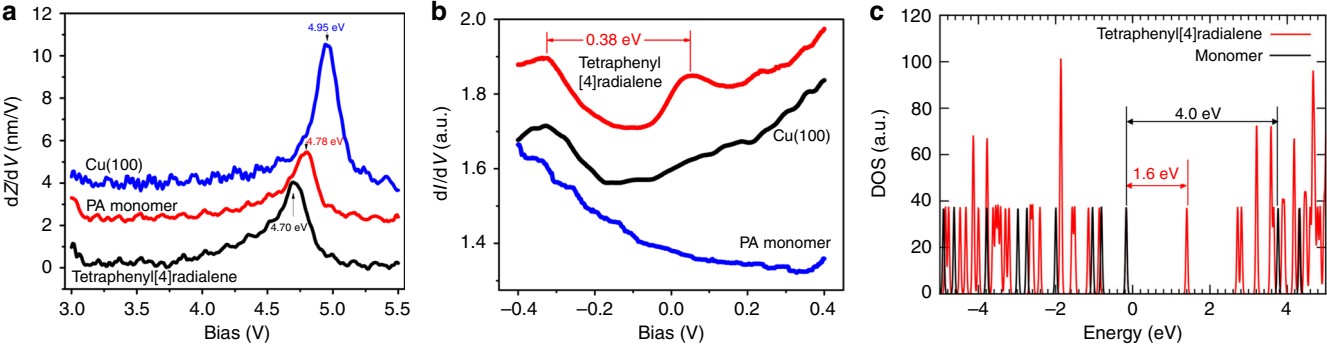

**Fig. 7** Spectra measurements for the clean Cu(100) surface, single PA molecule and the tetramer. d$Z$/d$V$ (**a**) and d$I$/d$V$ (**b**) spectra acquired on bare Cu (100), isolated PA monomers and newly formed tetraphenyl[4]radialene products. **c** The calculated HOMO–LUMO levels for isolated PA monomer and tetraphenyl[4]radialene in vacuum

moment induced by [4]radialene. By comparing the PA monomers, the tetraphenyl[4]radialene molecule possess a larger conjugation structure and thus exhibit stronger dipole moment. This agree well with our experimental observations. Further d$I$/d$V$ measurement (Fig. 7b) for isolated tetraphenyl[4]radialene shows a reduced HOMO–LUMO gap about 0.38 eV, again suggesting the formation of a more conjugated structure. We calculated the HUMO–LUMO levels for the isolated phenylacetylene monomer and tetraphenyl[4]radialene (without Cu surface). As shown in Fig. 7c, the HOMO–LUMO gap reduces significantly from ~4 eV on the monomer to ~1.6 eV on tetraphenyl[4]radialene molecule. Although the theoretical and experimental values cannot be directly compared due the electronic effects of Cu surfaces, the general trend of forming more conjugated molecular structures are in good agreement.

**nc-AFM measurements**. As discussed above, we provide various indirect evidences, including the STM observations, spectrum measurement, tip manipulation operations, and DFT calculations, all of them point towards the same conclusion. In order to obtain the direct evidence, nc-AFM measurements have been carried out[42–44]. As shown in Supplementary Figure 7, two bonds are observed at the center of the tetramer in the nc-AFM image, and we attribute them to the four-membered ring (see Supplementary Figure 7 for details). Self-assembly and metal–organic complex could not lead to such features. It is therefore reasonable to conclude that the [4]radialene has been successfully synthesized on the Cu(100) surface with our strategy.

## Discussion

Combining high-resolution STM experiments and first-principles DFT calculations, we demonstrate for the first time, that the tetraphenyl[4]radialene can be selectively produced from four phenylacetylene molecules at low temperatures on Cu(100) surfaces. Such a tetramerization reaction proceeds via a cooperative [1 + 1 + 1 + 1] cycloaddition mechanism, in which the terminal hydrogen atoms undergo inter-molecular tautomerization and bond to the carbon atoms that directly connect to the phenyl groups. Almost at the same time, the resulted carbon radicals combine with each other to form the core of a tetraphenyl[4] radialene molecule. The feasibility of this reaction pathway is dominated by the lattice symmetry and the electronic structures of Cu(100) surface. To be more specific, prior to the cycloaddition, the Cu(100) surface plays an important role to guide the molecular assembly of four phenylacetylene molecules to form a C–H/π bonded tetramer. Once such self-assembly structure is obtained, the Cu(100) surface serves an active catalyst that allows the

cooperative hydrogen tautomerization and C–C bond formation reactions to occur at low temperatures (before the molecular self-assembly is destroyed). In summary, this work presents not only a new and feasible synthetic method to selectively produce [4] radialenes, but also opens a new avenue to the C–C bond coupling reactions guided by the surface-induced assembly.

## Methods

**Sample preparation and STM experiments**. Experiments were conducted with a home-built variable temperature STM, whose temperature can vary from 10 to 300 K, and the base pressure is better than $1 \times 10^{-10}$ mbar. Phenylacetylene molecules were purified by the standard freeze-pump-thaw process, and the Cu (100) surface was cleaned by standard argon sputtering-annealing cycles. The Cu (100) substrate was kept at around 20 K during the molecular deposition. The tetramers are formed by subsequently annealing the sample to 200 K. A commercial Pt–Ir tip was prepared by gentle field emission at a clean Cu(100) sample. The bias voltage was applied on the sample during the STM observations. All the STM observations were taken at 10 K.

**DFT calculations**. The theoretical study was carried out in the framework of DFT using the Vienna ab initio Simulation Package[45]. In all calculations, the electron–ion interactions were described using the projected augmented-wave method[46]. The exchange-correlation energy was calculated with the general gradient approximation functionals of Perdew–Burke–Ernzerhof[47]. The plane-wave expansion was stopped at an energy cutoff of 400 eV. The van der waals interactions between the molecules and copper surfaces were corrected with a non-local correlation functional proposed by Dion[48]. The optimized lattice constant of Cu was 3.65 Å, which is in good agreement with previous theoretical values[48]. The Cu (100) surface was modeled with a (9 × 9) periodic slab consisting of four atomic layers. A vacuum of 20 Å was adopted to avoid the periodic image interactions normal to the surface. In all cases, the bottom two layers of Cu atoms were fixed at the bulk positions. The transition states were determined with the CI-NEB calculations, in which five structural images were inserted between the initial and the final states[49]. In all calculations, the reciprocal space was sampled with a 2 × 2 × 1 gamma centered Monkhorst–Pack grid[50]. The convergence thresholds of atomic forces for structure optimization and CI-NEB calculations were 0.01 eV/Å and 0.03 eV/Å, respectively. The STM simulation was performed with the Tersoff Hamman method using the bSKAN code[51].

**Data availability**. The data that support the findings of this study are available from the article and Supplementary Information files, or from the corresponding author upon reasonable request.

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

## Acknowledgements

We thank Prof. Klaus Müllen for helpful discussions. We acknowledge the Collaborative Innovation Center of Suzhou Nano Science & Technology and the Priority Academic Program Development of Jiangsu Higher Education Institutions. A portion of this work was conducted at the Center for Nanophase Materials Sciences, which is a DOE Office of Science User Facility. This work was supported by the National Science Foundation of China (21622306, 91545127, 21790053, 21771134, and 11574095), the Natural Science Foundation of Jiangsu Province (BK20140305, BK20150305), and the Major State Basic Research Development Program of China (2017YFA0205002, 2017YFA0204800, and 2014CB932600).

## Author contributions

Q.L. and M.H.P. designed the project and performed the STM measurements. H.P.L., Y.Y.L. and M.F.-C. carried out the DFT calculations and STM simulations. J.Z.G., Y.Y.L., H.P.L. and L.F.C. contributed to the interpretation of the data. All the authors contribute in the data analysis and the manuscript writing.

## Additional information

**Competing interests:** The authors declare no competing interests.

