## [Peer Review File · Nature Communications]

Reviewers' comments:

Reviewer #1 (Remarks to the Author):

The paper entitled “Self-assembly directed one-step synthesis of [4]radialene on Cu(100) surfaces”, by Pan et al., describes the funny phenomena on a copper catalyzed synthesis of [4]radialene via the cyclotetramerization of pheylacetylene upon thermal activation. Although the paper seems to be well written, this reviewer thinks this paper should not be accepted for publication under current form. It requires the major revision to reflect the reviewer’s comments, and to answer the doubtful points.

1. The authors shows the many “indirect” evidence of the four-membered ring of radialene such as STM images & simulation(Fig.3 b, c), tip manipulation (Fig. 5), and the mechanism by DFT calculation (Fig. 4) because the STM resolution can’t produce the real imaging of four-membered ring. The authors should show the “direct” evidence of this bond by doing the measurements of noncontact AFM.
2. With regards to the DFT calculation, the authors show the reaction pathway (Fig. 4) based on the hypothesis of “intermolecular” hydrogen transfer (tautomerization) triggered formation of four-membered rings. The word “inter-molecular hydrogen tautomerization” appears many times in abstract, manuscript and conclusions. However the reviewer feels doubtful this hypothesis because the “intramolecular” tautomerization can take place on annealing at 200 K because the hydrogen can be extracted by Cu surface to form carbene radical. The intramolecular tautomerization should be more favorable than intermolecular one. Additionally, the authors show no experimental evidence of “intermolecular” tautomerization. Why did the authors take only “intermolecular” tautomerization into account? The authors should perform the DFT calculation taking into account of the hypothesis on “intramolecular” tautomerization, and should make comparison. Authors should modify the words of “inter-molecular hydrogen tautomerization” to “hydrogen tautomerization” unless clear, and should add the comment in the main text.
3. Additionally regards to the DFT calculation, the authors adopt the hypothesis in which the C-C bonds form via one-step reaction from the carbon radicals of tetramer. However there is a possibility of sequential step-by-step radical coupling, such as one combines into two, three and four. The authors should consider not only one-step reactions but also the sequential reactions. The authors should reflect the comment in the manuscript.

4. The produced [4]radialene on Cu(100) should have the surface chirality. The righ-handed- and left-handed enantiomers should be identified in the STM images. The STM image of Fig 3b (at -100mV bias) seems to be asymmetry from the rectangular base (four-membered ring). Can the authors distinguish the isomers? The authors should add the comment on the manuscript.
5. The authors used the word “[2+2+2+2]” many times in the manuscript. However, in general, this means the number of carbon atoms which involve in the bond formation in the field of organic chemistry. In this case, four membered-ring is created from each carbon atom of four phenyl acetylene. Therefore, the reaction type should be [1+1+1+1]. The authors should check it.
6. On page 8, line 11, DFT and CI-NEB calculations, “From this meta-state to the final state (FS), the energy barrier is only 0.18 eV.”: “final state (FS)” should be mistake. May be the correct one is “TS2”.
7. On page 6, line 16, “As seen, the calculated protrusion-protrusion distance is around 1.60 Angstrom.”: However there is no indication of 1.60 Angstrom in Fig 3c. The authors should add it.
8. Fig. 5 caption: There is no explanation on the tip manipulations. The authors should add the detail operation. What do the figures of a, b, c mean?

Reviewer #2 (Remarks to the Author):

The authors report the on-surface synthesis of a [4]-radialene complex from molecular precursors equipped with alkyne groups.

The manuscript is well written and a nice combination of scanning tunneling microscopy and density functional theory calculations.

The synthesis of [4]-radialene is beyond any doubts of interest for the chemistry community.

However, the main conclusion of the manuscript, i.e. the synthesis of [4]-radialene, should be more elaborated. Thus, I recommend publication after my major criticism is addressed.

Major criticism:

- 1.- The synthesis of [4]-radialene is a hypotheis, but not fully proven. Since I like this study, I

would suggest the authors any of the following approaches:

* DFT of another paths of reaction. Is it really possible to have only [4]-radialene? Really? The review from Klappenberger et al. illustrates many competing mechanisms with alkyne chemistry on surfaces.

* Could the authors provide scanning tunneling spectroscopy and compare to a DFT calculation, to assess the identification of the [4]radialene.

* If the species are planar, best option could be non-contact atomic force microscopy.

I think that with any of these approaches, the study will be reinforced. I understand that STM+DFT is the state-of-the-art approach (provided there is not nc-AFM available), but even within this approach, STS and further DFT simulations are necessary.

Minor comments:

1.- High resolution image of Figure 1a is convenient.

2.- What are the white and black spots in Figure 1b?

3.- What is the role of entropy? Or in other words, how are thermodynamics taken into account into the calculations?

4.- Lateral manipulation of supramolecules are also possible with STM-tip...thus, this is a weak argumentation for covalent bonding, though frequently found in the community. I would suggest rewording to indication of strong coupling.

In summary, this study has the potential to be of interest for the chemistry community, though preliminar for publication.

Reviewer #3 (Remarks to the Author):

The manuscript of Li et al. reports a combined experimental and theoretical study of radialene molecules adsorbed on a single-crystal Cu(100) surface at ultrahigh vacuum conditions and cryogenic temperatures (below 120 K). The work reports experimental scanning tunneling microscopy data as well as results from first-principles calculations using density functional theory methods.

Li. et al claim thermally induced, selective, and on-surface catalyzed synthesis of [4]radialene molecules obtained from phenylacetylene molecular precursors via a [2+2+2+2] mechanism.

The topic of on-surface chemistry is timely and important.

The claims are not fully supported by the presented data/results.

The reported synthesis of [4]radialene molecules in such small

amounts of only sub-monolayer coverages on a single-crystal surface at cryogenic temperature seems to be yet another molecule-on-a-surface work.

I doubt that the work reported here will be considered extremely important by experts in the field of synthetic organic chemistry as hoped by the authors.

The appeal to researchers from other fields is moderate, as well.

In summary, I recommend submission to a more specialized journal after adding complementary experimental evidence that sufficiently support the all claims in this work.

In detail, my concerns are as follows:

1. I see no direct experimental evidence for the claimed role of the terminal hydrogen atom in the proposed reaction scheme.

Plain comparison of experimental STM topographic images with DFT results alone does not allow to judge on this crucial point.

In order to substantiate the claimed cycloaddition reaction, as proposed in the manuscript, releasing no hydrogen atoms while synthesizing every single radialene molecule, further experiments are required: for instance, temperature-programmed desorption experiments are expected to lack significant hydrogen desorption at the proposed reaction temperature.

2. The authors claim carbon-radical intermediates (page 5) in the reaction scheme without presenting any experimental evidence for this claim.

3. The presented tip-manipulation experiments (Fig. 5) are insufficient to prove the claimed nature of the molecular reaction product. After setting the right parameters, pick-up by the STM tip commonly works with almost any weakly adsorbed small molecule as well as group of molecules.

In particular, it would be interesting to see the result of trying to put the molecule back on the Cu substrate again.

Reviewers' comments:

Reviewer #1 (Remarks to the Author):

The paper entitled “Self-assembly directed one-step synthesis of [4]radialene on Cu(100) surfaces”, by Pan et al., describes the funny phenomena on a copper catalyzed synthesis of [4]radialene via the cyclotetramerization of phenylacetylene upon thermal activation. Although the paper seems to be well written, this reviewer thinks this paper should not be accepted for publication under current form. It requires the major revision to reflect the reviewer’s comments, and to answer the doubtful points.

1. The authors shows the many “indirect” evidence of the four-membered ring of radialene such as STM images & simulation(Fig.3 b, c), tip manipulation (Fig. 5), and the mechanism by DFT calculation (Fig. 4) because the STM resolution can’t produce the real imaging of four-membered ring. The authors should show the “direct” evidence of this bond by doing the measurements of noncontact AFM.

Author Reply: Thanks for the reviewer’s constructive comments. We agree that nc-AFM could provide more direct evidence, despite all our “indirect evidence” points towards the same conclusion that [4]radialenes are formed. The difficulty of the experiments is one needs to meet the requirement for nc-AFM imaging and low-temperature PA molecule adsorption simultaneously. Fortunately, with collaborated with Prof. Xiaohui Qiu’s group (we thus change the authors list accordingly), we reproduced successfully the [4]radialene molecules in their system and further characterized by noncontact AFM, and obtained some AFM results according to the referee’s suggestion.

Figure R1(b) shows the typical nc-AFM image for a single [4]radialene, acquired with a STM image shown in Figure R1(a) at the same location. At the center of the [4]radialene molecule (marked by yellow dashed rectangle), nc-AFM image shows the sign for *two bonds* of four-membered ring. ***Self-assembly and metal-organic complex could not lead to such features.***

In addition, one may notice that each phenyl ring of the [4]radialene can only partially be resolved in the nc-AFM image, as shown in Figure R1(b). It can be explained by the unplanar configurations of the [4]radialenes (similar conclusion can be found in *J. Phys. Chem. C* 2018, 122, 4997). Nc-AFM technique can only images the topmost chemical bonds for an unplanar molecule. In order to make it more clear, the nc-AFM simulations based on out-of-plane state of the [4]radialene (Figure R1(c)) are carried out. As shown in Figure R1(d), the simulated

nc-AFM image only partially resolve the phenyl rings. The results of nc-AFM measurements have been added to the supporting information as Fig. S7.

Figure R1. A typical nc-AFM image (b) for a single [4]radialene, acquired with a STM image at the same location (a); (c) the front and top views of a [4]radialene molecule with unplanar configuration, in which four phenyl-rings tilt slightly out-of-plane and the four-membered ring sinks down. (d) The simulated nc-AFM image based on the structure shown in (c).

Besides that, we have carried out tunneling spectroscopic measurement (including dZ/dV , dI/dV) at single molecule scale, which provide alternative evidence for the formation of the four-membered ring of radialene.

We added Figure 6 and related discussion in the revised version:

Page 11, line 4 add “Finally, we performed the dZ/dV (represents the image potential, and related to the local density of states) and dI/dV measurements for the clean Cu(100) surface, single PA molecule and the tetramer (Figure 6). dZ/dV spectroscopy (Fig. 6a) shows the Stark-shifted, first image-potential state associated with phenylacetylene monomer and tetraphenyl[4]radialene, compared to bare Cu(100) surface. It is believed that a molecular dipole moment induced by molecular adsorption, lowers the surface work function, which cause to the observed shift of image-potential state. For isolated tetraphenyl[4]radialene molecule, it induces 0.08 eV more energy downshift than that of isolated PA monomers, suggesting a stronger dipole moment induced by [4]radialene. By comparing the PA monomers, the tetraphenyl[4]radialene molecule possess a larger conjugation structure and thus exhibit stronger dipole moment. This agree well with our experimental observations. Further dI/dV measurement (Fig. 6b) for isolated tetraphenyl[4]radialene shows a reduced HOMO-LUMO gap about 0.38 eV, again suggesting the formation of a more conjugated structure. We calculated the HUMO-LUMO levels for the isolated phenylacetylene monomer and tetraphenyl[4]radialene (without Cu surface). As shown in Fig. 6c, the HOMO-LUMO gap reduces significantly from ~ 4 eV on the monomer to ~ 1.6 eV on tetraphenyl[4]radialene molecule. Although the theoretical and experimental values cannot be directly compared due the electronic effects of Cu surfaces, the general trend of forming more conjugated molecular structures are in nice agreement.”

Add Figure R2 (Figure 6 in the manuscript).

Figure R2. dZ/dV (a) and dI/dV (b) spectrum acquired on bare Cu(100), isolated PA monomers and newly formed tetraphenyl[4]radialene products. (c) the calculated HOMO-LUMO levels for isolated PA monomer and tetraphenyl[4]radialene in vacuum.

As discussed above, we provide various indirect evidences, including the STM observations, spectrum measurement, tip manipulation operations and DFT calculations; and the direct evidence (nc-AFM). All of them point towards the same conclusion. It is therefore reasonable to conclusion that the [4]radialene has been successfully synthesized on the Cu(100) surface.

2. With regards to the DFT calculation, the authors show the reaction pathway (Fig. 4) based on the hypothesis of “intermolecular” hydrogen transfer (tautomerization) triggered formation of four-membered rings. The word “inter-molecular hydrogen tautomerization” appears many times in abstract, manuscript and conclusions. However the reviewer feels doubtful this hypothesis because the “intramolecular” tautomerization can take place on annealing at 200 K because the hydrogen can be extracted by Cu surface to form carbene radical. The intramolecular tautomerization should be more favorable than intermolecular one. Additionally, the authors show no experimental evidence of “intermolecular” tautomerization. Why did the authors take only “intermolecular” tautomerization into account? The authors should perform the DFT calculation taking into account of the hypothesis on “intramolecular” tautomerization, and should make comparison. Authors should modify the words of “inter-molecular hydrogen tautomerization” to “hydrogen tautomerization” unless clear, and should add the comment in the main text.

Author Reply: We thank the reviewer for the constructive advices. Additional DFT calculations have been performed to study the intra-molecular hydrogen transfer followed by the cyclization. The corresponding energy barrier of such reaction pathway is higher than the inter-molecular hydrogen transfer. The results of new calculations have been added to the supporting information as Fig. S5 and Fig. S6 (also see Fig. R3 and R4). We thus believe the [4]radialenes are formed via the inter-molecular hydrogen transfer.

Figure R3. The reaction pathway of intra-molecular hydrogen transfer, and subsequent C-C bond coupling of two phenylacetylene molecules. The calculated energy barrier is 1.55 eV.

Figure R4. The reaction pathway of intra-molecular hydrogen transfer, and subsequent C-C bond coupling of four phenylacetylene molecules to produce the tetraphenyl[4]radialene. The calculated energy barrier is 2.59 eV.

3. Additionally regards to the DFT calculation, the authors adopt the hypothesis in which the C-C bonds form via one-step reaction from the carbon radicals of tetramer. However there is a possibility of sequential step-by-step radical coupling, such as one combines into two, three and four. The authors should consider not only one-step reactions but also the sequential reactions. The authors should reflect the comment in the manuscript.

Author Reply: We are really thankful to the reviewer for this question that has given us the chance to clarify even more. Based on our understanding, the concerted reaction mechanism requires the lowest activation, *i.e.* reaction barrier for each monomer is about 0.34 eV, which is close to their diffusion barriers. In order to compare the difference between the concerted and non-concerted reaction pathways. We have performed additional DFT calculations to study the case of dimer formation. As shown in Fig. S5 (also Fig. R3), the energy barrier is 1.55 eV, which is too high compared to the experimental observations. In addition, the dimer should diffuse slower than the monomer, experimentally, we did not see any dimer-like features. We thus think that the concerted reaction mechanism is more convincing.

4. The produced [4]radialene on Cu(100) should have the surface chirality. The right-handed- and left-handed enantiomers should be identified in the STM images. The STM image of Fig 3b (at -100mV bias) seems to be asymmetry from the rectangular base (four-membered ring). Can the authors distinguish the isomers? The authors should add the comment on the manuscript.

Author Reply: This is very good comment. These two isomers with different chirality can only be distinguished by clearly imaging the orientation of the C=C bonds connected with

four-membered ring. Unfortunately, the orientation of the C=C bond is hardly seen from STM image. See Fig. R5(a), we cannot distinguish it as the right-handed (b) or left-handed (c) enantiomers.

Figure R5. STM image (a), the models of the right-handed (b) or left-handed (c) enantiomers.

5. The authors used the word “[2+2+2+2]” many times in the manuscript. However, in general, this means the number of carbon atoms which involve in the bond formation in the field of organic chemistry. In this case, four membered-ring is created from each carbon atom of four phenyl acetylene. Therefore, the reaction type should be [1+1+1+1]. The authors should check it.

Author Reply: We adopt this reviewer’s suggestion, and correct it in the revised version.

6. On page 8, line 11, DFT and CI-NEB calculations, “From this meta-state to the final state (FS), the energy barrier is only 0.18 eV.”: “final state (FS)” should be mistake. May be the correct one is “TS2”.

Author Reply: We adopt this reviewer’s suggestion, and correct it in the revised version.

7. On page 6, line 16, “As seen, the calculated protrusion-protrusion distance is around 1.60 Angstrom.”: However there is no indication of 1.60 Angstrom in Fig 3c. The authors should add it.

Author Reply: Thanks for the careful review, and we have added the distance in Figure 3c.

8. Fig. 5 caption: There is no explanation on the tip manipulations. The authors should add the detail operation. What do the figures of a, b, c mean?

Author Reply: We adopt this reviewer’s suggestion, and add in the revised version:

Page 10, line 15 add “To conduct the tip manipulation, we applied a negative voltage pulse of -3.2 V between STM tip and molecules (details of the manipulation methods can be found in ref. 32). We choose a region where three tetramers are observed (Fig. 5a). After the pulse is applied, the tetramer located at the top left corner disappears (Fig. 5b). Further tip manipulation results in that the STM tip picks up one more tetramer, as seen in Fig. 5c.”

Reviewer #2 (Remarks to the Author):

The authors report the on-surface synthesis of a [4]-radialene complex from molecular precursors equipped with alkyne groups.

The manuscript is well written and a nice combination of scanning tunneling microscopy and density functional theory calculations. The synthesis of [4]-radialene is beyond any doubts of interest for the chemistry community.

Author Reply: Thanks for the reviewer’s kind evaluations.

However, the main conclusion of the manuscript, i.e. the synthesis of [4]-radialene, should be more elaborated. Thus, I recommend publication after my major criticism is addressed.

Major criticism:

1.- The synthesis of [4]-radialene is a hypothesis, but not fully proven. Since I like this study, I would suggest the authors any of the following approaches: DFT of another paths of reaction. Is it really possible to have only [4]-radialene? Really? The review from Klappenberger et al. illustrates many competing mechanisms with alkyne chemistry on surfaces.

Author Reply: Thanks for the reviewer’s constructive suggestions, and we do agree that alkyne chemistry on surfaces is quite complicate (see the review by Klappenberger *et al.* (ref. 20)). Actually, we have considered other candidate structures for the tetramer product, such as self-assembly (Figure S4) and intra-molecular hydrogen transfer (Figure S5 and S6, also see Fig. R3 and R4). All of them exhibit a larger size or higher reaction barrier than the proposed mechanism. In addition, we have considered the Cu complex product, and the 8-membered product shown below (Fig. R6). They are much less stable than the tetraphenyl[4]radialene molecule.

Figure R6. Cu complex product, and the 8-membered product

We have added Figure S5 and S6 in the supporting informations.

** Could the authors provide scanning tunneling spectroscopy and compare to a DFT calculation, to assess the identification of the [4]radialene.*

Author Reply: As suggested by reviewer, we have carried out tunneling spectroscopic measurement (including dZ/dV , dI/dV) at single molecule scale, which provide alternative “indirect evidence” for the formation of the four-membered ring of radialene. The spectroscopy results are compared with the our DFT calculations

We added Figure 6 and related discussion in the revised version:

Page 11, line 4 add “Finally, we performed the dZ/dV (represents the image potential, and related to the local density of states) and dI/dV measurements for the clean Cu(100) surface, single PA molecule and the tetramer (Figure 6). dZ/dV spectroscopy (Fig. 6a) shows the Stark-shifted, first image-potential state associated with phenylacetylene monomer and tetraphenyl[4]radialene, compared to bare Cu(100) surface. It is believed that a molecular dipole moment induced by molecular adsorption, lowers the surface work function, which cause to the observed shift of image-potential state. For isolated tetraphenyl[4]radialene molecule, it induces 0.08 eV more energy downshift than that of isolated PA monomers, suggesting a stronger dipole moment induced by [4]radialene. By comparing the PA monomers, the tetraphenyl[4]radialene molecule possess a larger conjugation structure and thus exhibit stronger dipole moment. This agree well with our experimental observations. Further dI/dV measurement (Fig. 6b) for isolated tetraphenyl[4]radialene shows a reduced HOMO-LUMO gap about 0.38 eV, again suggesting the formation of a more conjugated structure. We calculated the HUMO-LUMO levels for the isolated phenylacetylene monomer and tetraphenyl[4]radialene (without Cu surface). As shown in Fig. 6c, the HOMO-LUMO gap reduces significantly from ~ 4 eV on the monomer to ~ 1.6 eV on tetraphenyl[4]radialene

molecule. Although the theoretical and experimental values cannot be directly compared due to the electronic effects of Cu surfaces, the general trend of forming more conjugated molecular structures are in nice agreement.”

** If the species are planar, best option could be non-contact atomic force microscopy. I think that with any of these approaches, the study will be reinforced. I understand that STM+DFT is the state-of-the-art approach (provided there is not nc-AFM available), but even within this approach, STS and further DFT simulations are necessary.*

Author Reply: Thanks for the reviewer’s constructive comments. We agree that nc-AFM could provide more direct evidence, despite all our “indirect evidence” points towards the same conclusion that [4]radialenes are formed. The difficulty of the experiments is one needs to meet the requirement for nc-AFM imaging and low-temperature PA molecule adsorption simultaneously. Fortunately, with collaborated with Prof. Xiaohui Qiu’s group (we thus change the authorlist accordingly), we obtained some AFM results according to the referee’s suggestion.

The details of the nc-AFM results can be seen in the reply for question 1 of reviewer 1.

Minor comments:

1.- High resolution image of Figure 1a is convenient.

Author Reply: We replaced Figure 1a with higher resolution image.

2.- What are the white and black spots in Figure 1b?

Author Reply: The white spots in Fig. 1b are the PA monomers. The black spots could be some surface defects, such as adsorbates.

3.- What is the role of entropy? Or in other words, how are thermodynamics taken into account into the calculations?

Author Reply: Thanks for the comments. The total energy in DFT calculations are the Helmholtz free energies. It is represented by the equation: $E = U - TS$, in which U, T, S are referred to as the internal energy, temperature, and entropy, respectively. In the standard DFT calculations, the temperature effects are not included, which means that the energies obtained with DFT is more reliable when it is used to interpret the low-temperature experimental observations. In this work, the formation of [4]radialene takes place below 77 K. The

energies from DFT calculations should be accurate enough to account for the reaction barriers. For the role of entropy, we do not think it plays a significant role for two reasons: Firstly, the molecules are adsorbed on the Cu(100) surface, the degree of freedom of molecular vibrations is depressed due to the surface-adsorbate interactions. Secondly, the experiments were conducted at very low temperatures, therefore, the contribution from TS should be negligible. The thermodynamics can be used to exclude the possibility of self-assembly products. Because experimentally, our previous studies have indicated that depositing phenylacetylene molecules onto the same Cu(100) surface held at 120 K leads to the formation of one dimensional polymers (Sci. Rep. 2013, 3, 2102). Thermodynamically, one should not obtain self-assembly structures after annealing the phenylacetylene decorated Cu(100) surfaces at a temperature much higher than 120 K.

4.- Lateral manipulation of supramolecules are also possible with STM-tip...thus, this is a weak argumentation for covalent bonding, though frequently found in the community. I would suggest rewording to indication of strong coupling.

Author Reply: Thanks for the comments. We thought the monomers are connected covalently because different behaviors are observed after tip manipulations.

According to our previous study, we have prepared phenylacetylene hexamers on a Au(111) surface, which are known to be formed by six pristine phenylacetylene molecules via non-covalent C-H/ π interactions. By means of the same tip manipulation method, it is shown that only one phenylacetylene monomer is picked up by the STM tip (see the supporting information in ACS Nano 2012, 6, 566 for details). For the tetramers on the Cu(100) surface, however, the entire tetramer is picked up by the STM tip. We thus believe the tetramers are the reaction products of cycloaddition.

In the revised version, we lowered the tune:

Page 10, Line 20 “This phenomenon clearly indicates that each tetramer structure is composed of four monomers which are connected by strong attractive interactions, *e.g.* the covalent bonds.”

Page 10, Line 26 “The different results of tip manipulation analysis for hexamers on Au(111) and tetramers on Cu(100) then suggest that the tetramers are not self-assembly structures, but the reaction products of cycloaddition.”

In summary, this study has the potential to be of interest for the chemistry community, though preliminar for publication.

Author Reply: We hope the revised version of the manuscript answers all the questions the reviewer has raised, and the manuscript can eventually be published.

Reviewer #3 (Remarks to the Author):

The manuscript of Li et al. reports a combined experimental and theoretical study of radialene molecules adsorbed on a single-crystal Cu(100) surface at ultrahigh vacuum conditions and cryogenic temperatures (below 120 K). The work reports experimental scanning tunneling microscopy data as well as results from first-principles calculations using density functional theory methods. Li. et al claim thermally induced, selective, and on-surface catalyzed synthesis of [4]radialene molecules obtained from phenylacetylene molecular precursors via a [2+2+2+2] mechanism. The topic of on-surface chemistry is timely and important. The claims are not fully supported by the presented data/results. The reported synthesis of [4]radialene molecules in such small amounts of only sub-monolayer coverages on a single-crystal surface at cryogenic temperature seems to be yet another molecule-on-a-surface work. I doubt that the work reported here will be considered extremely important by experts in the field of synthetic organic chemistry as hoped by the authors. The appeal to researchers from other fields is moderate, as well.

Author Reply: There probably some misunderstanding of the manuscript. This work does reported a novel strategy for the synthesis of [4]radialene on surfaces. However, this is not a purely synthetic organic chemistry work. The real highlight of the manuscript is we found that the reaction directions/products are completely different (even with the same precursor and catalysts, see *Sci. Rep.* 2013, 3, 2102) when molecular assembly was prohibited. Of more fundamental advances, we give an example showing the significance of tuning the reaction pathways with chemical kinetics.

In detail, my concerns are as follows:

1. I see no direct experimental evidence for the claimed role of the terminal hydrogen atom in the proposed reaction scheme. Plain comparison of experimental STM topographic images with DFT results alone does not allow to judge on this crucial point. In order to substantiate the claimed cycloaddition reaction, as proposed in the manuscript, releasing no hydrogen atoms while synthesizing every single radialene molecule, further experiments are required: for instance, temperature-programmed desorption experiments are expected to lack significant hydrogen desorption at the proposed reaction temperature.

In summary, I recommend submission to a more specialized journal after adding complementary experimental evidence that sufficiently support the all claims in this work.

Author Reply: We thanks the reviewer for the nice comments. Similar suggestions are also raised by reviewer 1 and 2.

We have performed the nc-AFM and spectroscopy measurement accordingly. The details can be found in the reply for the question 1 of the reviewer 1.

2. The authors claim carbon-radical intermediates (page 5) in the reaction scheme without presenting any experimental evidence for this claim.

Author Reply: We thank the reviewer for this comment. In our DFT calculations, the energy barrier from carbon-radical intermediates (MS) to the final products (FS) is only 0.18 eV (shown in Fig. 4), which is even lower than that of the diffusion barrier of PA molecules. Such low energy barrier suggests that the transfer from MS to FS is much faster than molecular diffusions. Taking into account that the intermediate states of molecular diffusions cannot be imaged by STM due their short life-time. The missing STM images of such MS can also be interpreted by its short life-time.

3. The presented tip-manipulation experiments (Fig. 5) are insufficient to prove the claimed nature of the molecular reaction product. After setting the right parameters, pick-up by the STM tip commonly works with almost any weakly adsorbed small molecule as well as group of molecules.

Author Reply: Thanks for the comments. We thought the monomers are connected covalently because different behaviors are observed after tip manipulations.

According to our previous study, we have prepared phenylacetylene hexamers on a Au(111) surface, which are known to be formed by six pristine phenylacetylene molecules via non-covalent C-H/ π interactions. By means of the same tip manipulation method, it is shown that only one phenylacetylene monomer is picked up by the STM tip (see the supporting information in ACS Nano 2012, 6, 566 for details). For the tetramers on the Cu(100) surface, however, the entire tetramer is picked up by the STM tip. We thus believe the tetramers are the reaction products of cycloaddition.

In particular, it would be interesting to see the result of trying to put the molecule back on the Cu substrate again.

Author Reply: Thanks for the comments. However, we have failed to put the molecule back on the Cu substrate again.

Reviewers' Comments:

Reviewer #1 (Remarks to the Author):

This reviewer feels the revised manuscript has been correctly modified according to the reviewer's comments, and also the authors well replied to the technical questions that the reviewer pointed out. I respect author's much efforts. I think this paper should be accepted for publication.

Reviewer #2 (Remarks to the Author):

The authors have answered all comments from referees and the manuscript has notably been improved.

Thus I would recommend publication as it is.